

# Testing pterosaur ingroup relationships through broader sampling of avemetatarsalian taxa and characters and a range of phylogenetic analysis techniques

Matthew G. Baron[1,2]

[1] BPP University, London, UK
[2] Christ's College, University of Cambridge, Cambridge, UK

Corresponding author
Matthew G. Baron,
m.baron2@my.bpp.com

## ABSTRACT

The pterosaurs first appear in the fossil record in the middle of the Late Triassic. Their earliest representatives are known from Northern Hemisphere localities but, by the end of the Jurassic Period, this clade of flying reptiles achieved a global distribution, as well as high levels of diversity and disparity. Our understanding of early pterosaur evolution and the fundamental interrelationships within Pterosauria has improved dramatically in recent decades. However, there is still debate about how the various pterosaur subgroups relate to one another and about which taxa comprise these. Many recent phylogenetic analyses, while sampling well from among the known Triassic and Early Jurassic pterosaurs, have not included many non-pterosaurian ornithodirans or other avemetatarsalians. Given the close relationship between these groups of archosaurs, the omission of other ornithodirans and avemetatarsalians has the potential to adversely affect the results of phylogenetic analyses, in terms of character optimisation and ingroup relationships recovered. This study has addressed this issue and tests the relationships between the early diverging pterosaur taxa following the addition of avemetatarsalian taxa and anatomical characters to an existing early pterosaur dataset. This study has, for the first time, included taxa that represent the aphanosaurs, lagerpetids, silesaurids and dinosaurs, in addition to early pterosaurs. Anatomical characters used in other recent studies of archosaurs and early dinosaurs have also been incorporated. By expanding the outgroup taxa and anatomical character coverage in this pterosaur dataset, better resolution between the taxa within certain early pterosaur subclades has been achieved and stronger support for some existing clades has been found; other purported clades of early pterosaurs have not been found in this analysis—for example there is no support for a monophyletic Eopterosauria or Eudimorphodontidae. Further support has been found for a sister-taxon relationship between *Peteinosaurus zambelli* and Macronychoptera, a clade here named Zambellisauria (clade nov.), as well as for a monophyletic and early diverging Preondactylia. Some analyses also support the existence of a clade that falls as sister-taxon to the zambellisaurs, here named Caviramidae (clade nov.). Furthermore, some support has been found for a monophyletic Austriadraconidae at the base of Pterosauria. Somewhat surprisingly, Lagerpetidae is recovered outside of Ornithodira *sensu stricto*, meaning that, based upon current definitions at least, pterosaurs fall within Dinosauromorpha in this analysis. However, fundamental

ornithodiran interrelationships were not the focus of this study and this particular result should be treated with caution for now. However, these results do further highlight the need for broader taxon and character sampling in phylogenetic analyses, and the effects of outgroup choice on determining ingroup relationships.

## INTRODUCTION

Pterosaurs were a diverse, disparate and highly specialised group of terrestrial reptiles that represent the oldest set of vertebrates currently understood to have achieved powered flight (*Benton, 1985*; *Unwin, 2003*; *Andres, 2006*; *Barrett et al., 2008*; *Andres, Clark & Xu, 2014*; *Britt et al., 2018*). Originating at some time in either the Early or Middle Triassic (*Nesbitt et al., 2017*), and first appearing in the fossil record in the middle of the Late Triassic (*Barrett et al., 2008*; *Bennett, 2013*), the pterosaurs went on to thrive throughout the Mesozoic Era as one of the dominant groups of land animals, lasting right up until the very end of the Cretaceous Period and achieving a global distribution (*Unwin, 2003*; *Dalla Vecchia, 2004*; *Unwin & Martill, 2007*; *Barrett et al., 2008*; *Kellner et al., 2019*).

The earliest pterosaurs were generally small bodied animals, with toothed upper and lower jaws and usually an elongated tail (*Padian, 1984*, *2008a*, *2008b*; *Hone & Benton, 2007*; *Bennett, 2007*, *2014*; *Kellner, 2015*; *Britt et al., 2018*; *Dalla Vecchia, 2010*, *2013*, *2019*). In addition, all known early pterosaurs appear to be fully capable of powered flight and, as yet, no transitional non-flying pterosaur taxa are known (though some specimens have been suggested to be exactly that—*Huene, 1914*). Later pterosaurs went on to achieve a broader, truly global, geographic range, as well as much larger body sizes and much more unusual and often unique features of anatomy (*Unwin & Bakhurina, 1994*, *1995*; *Unwin, 2001*, *2003*; *Dalla Vecchia et al., 2002*; *Barrett et al., 2008*; *Hone et al., 2012*; *Upchurch et al., 2015*; *Kellner et al., 2019*).

Within Pterosauria, which currently comprises the same set of taxa as the clade Pterosauromorpha (see, *Nesbitt, Desojo & Irmis, 2013*), there exists a number of distinct subgroups, many of which were already present in the Late Triassic. The many proposed subgroups within Pterosauria include the Eopterosauria, which is believed to comprise Preondactylia and Eudimorphodontoidea (*Andres, Clark & Xu, 2014*), and Macronychoptera, which comprises, inter alia, Dimorphodontidae, Anurognathidae and Pterodactyloidea (*Britt et al., 2018*; *Dalla Vecchia, 2019*) (Figs. 1A–1C).

The recent phylogenetic analysis by *Dalla Vecchia (2019)* suggested that the earliest diverging members of Pterosauria that were considered by that study were the Preondactylia—comprising *Preondactylus buffarinii* and *Austriadactylus cristatus*. This pairing of *Preondactylus buffarinii* and *Austriadactylus cristatus* concurs with the findings of *Andres, Clark & Xu (2014)*. However, the results of these two analyses differ in that in the trees produced by *Andres, Clark & Xu (2014)*, the Preondactylia form the sister-taxon to the clade comprising *Peteinosaurus zambellii* and Eudimorphodontoidea,

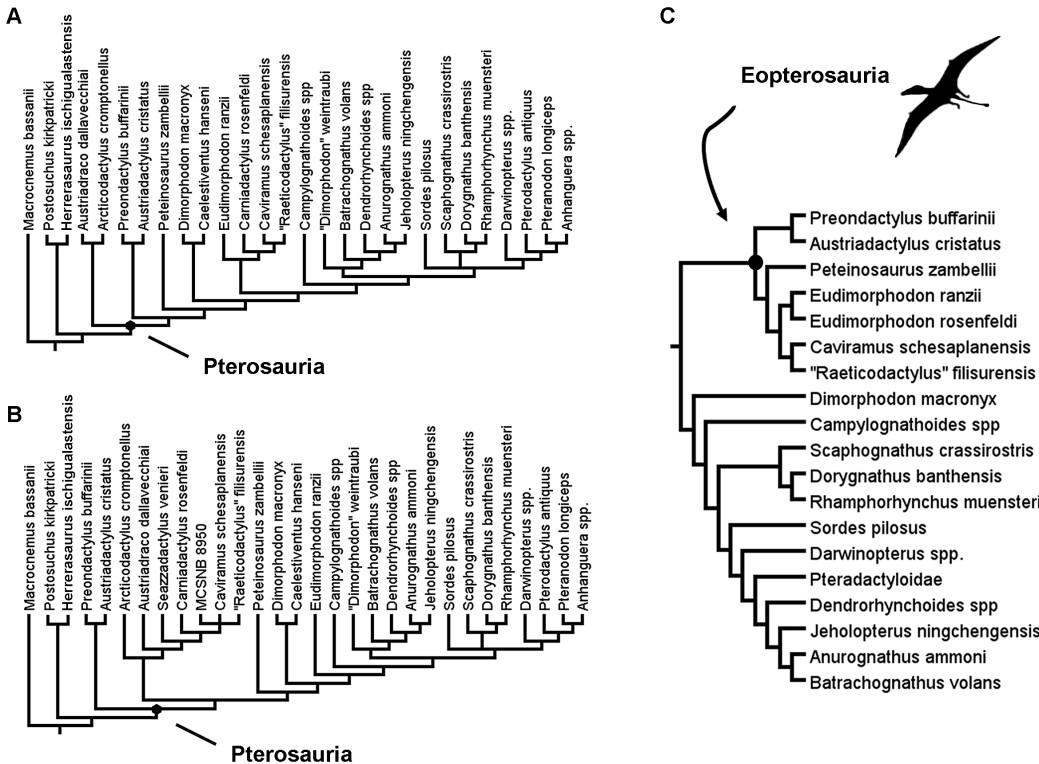

**Figure 1 Previous recent hypotheses of early pterosaur relationships.** (A) The results of the reduced taxon analysis by *Britt et al. (2018)*; (B) the results of the analysis by *Dalla Vecchia (2019)*; (C) the results of the analysis by *Andres, Clark & Xu (2014)*.

a topology also supported by *Upchurch et al. (2015)*. These clades together form Eopterosauria, a group not supported in the analysis by *Britt et al. (2018)* or *Dalla Vecchia (2019)*. Equally, in the analysis by *Dalla Vecchia (2019)* no support was found for the clade Eudimorphodontidae (sensu *Dalla Vecchia, 2014*). In this respect, the analyses of *Dalla Vecchia (2019)* and *Britt et al. (2018)* also differ from each other, despite using the same dataset and having only a very small number of differences in terms of the operational taxa included (see Fig. 1). This just highlights the relative instability of some of the early branching pterosaur taxa in phylogenetic analyses. *Dalla Vecchia (2019)* even admits that Bremer support values for many of the clades in his analysis were low.

A monophyletic Macronychoptera was found to be in a sister-taxa relationship with *Peteinosaurus zambellii* in the analysis of *Dalla Vecchia (2019)*, and this unnamed clade was found as sister-taxon to another unnamed group of pterosaurs containing a number of other Late Triassic forms (Fig. 1B). Unlike in the analysis by *Andres, Clark & Xu (2014)*, Dalla Vecchia recovers *Eudimorphodon ranzii* within Lonchognatha, and finds no evidence for a close relationship between *Eudimorphodon ranzii*, *Carniadactylus rosenfeldi* and *Arcticodactylus cromptonellus* (*contra Andres, Clark & Xu, 2014*) (Figs. 1A and 1B). Similarly, a close affinity between *Eudimorphodon ranzii* and Raeticodactylidae—which comprises *Raeticodactylus filisurensis* and *Caviramus schesaplanensis* according to *Andres, Clark & Xu (2014)*—was found by *Britt et al. (2018)* but not *Dalla Vecchia (2019)*

(Figs. 1A–1C). In the reduced strict consensus tree produced in the analysis by *Britt et al. (2018)*, the earliest diverging pterosaur clade contained a pair of taxa—*Austriadraco dallavecchiai* and *Arcticodactylus cromptonellus*—a similar result to that obtained by *Kellner (2015)*, who named this clade Austriadraconidae (see also, *Dalla Vecchia, 2009a, 2009b*). But again, this result differs from the analysis of both *Upchurch et al. (2015)* and *Dalla Vecchia (2019)*. In an analysis by *Codorniú et al. (2016)*, the results vary even more dramatically at the base of Pterosauira, with Anurognathidae diverging earlier within the trees recovered than either *Eudimorphodon ranzii* or *Austriadactylus cristatus*. *Codorniú et al. (2016)* also found evidence for a possible a *Dimorphodon macronyx* + *Peteinosaurus zambelli* sister-taxon relationship, which was not recovered in any of the other analyses discussed above.

It is clear from all of the differences observed between the various analyses discussed above that much work still needs to be done to resolve the interrelationships between the many early pterosaur taxa currently known. The ingroup relationships of the various early pterosaur clades are unstable, as are the interrelationships between clades. However, in all the studies discussed above, there is a potential problem with the analyses in that they perhaps do not include adequate sampling from without Pterosauria—that is, the lack of informative anatomical information from certain key outgroup taxa could be causing the poor resolution within Pterosauria.

In the analysis of *Britt et al. (2018)*, and of other studies that utilised the same data (*Codorniú et al., 2016*; *Dalla Vecchia, 2019*), the only other ornithodiran taxa to be included in the analysis as an outgroup taxon is the unusual hypercarnivore *Herrerasaurus ischigualastensis*. The taxon was presumably chosen as a representative of the Dinosauria, a clade supposedly closely related to Pterosauria. However, *H. ischigualastensis* is not necessarily the best representative of the 'basal' dinosaurian condition, being a very large predator that is quite distinct in terms of its anatomy to many, if not most, of the earliest dinosaurs (see, *Brusatte et al., 2010*; *Baron, Norman & Barrett, 2017a*). In fact, the position of this taxon has proved to be highly unstable in recent times (see, *Baron, Norman & Barrett, 2017b*; *Langer et al., 2017*; *Lee et al., 2019*; *Pacheco et al., 2019*) and belongs to a wider clade of Triassic hypercarnivores that may or may not fall within Dinosauria at all (see, *Baron & Williams, 2018*). Moreover, in the analyses by *Britt et al. (2018)* and *Dalla Vecchia (2019)*, *H. ischigualastensis* is recovered as the sister taxon to the rauisuchid paracrocodylomorph *Postosuchus kirkpatricki*, which might suggest that character optimisation outside of the pterosaurian lineage is somewhat confused and misleading. If the character distribution among taxa in this analysis was fairly reflective of the topology expected to be found for these taxa, *H. ischigualastensis* should, according to almost all modern phylogenetic hypotheses, fall closer to the pterosaurs than to *Postosuchus kirkpatricki*. This result, while not the key focus of any of the studies that recovered it, perhaps should have raised alarm bells in terms of what the data for taxa immediately at the base of and just outside of Pterosauria was like. The purpose of outgroup taxa is to reflect, as best as is possible, the 'basal' condition for the ingroup clade being studied—it is arguable that this is not the case in the analyses by *Britt et al. (2018)* and *Dalla Vecchia (2019)* and that these analyses fall short in this key respect.

This omission of important anatomical data may, in turn, be having a substantial adverse effect on the resolution of the ingroup relationships among the numerous pterosaur taxa included in the studies.

Also missing from the datasets is a range of other close pterosaur relatives, the anatomical characteristics of which are potentially even more helpful in determining the ancestral state of Pterosauria than *H. ischigualastensis* is when considered alone. Silesaurids, who along with dinosaurs form the dinosauromorph clade Dracohors (*Cau, 2018*) are omitted, as are the dinosauriforms known as the lagerpetids. Similarly, looking the other way along this particular branch of the archosaur group, the aphanosaurs, a group believed to form the sister-taxon to the ornithodirans (see *Nesbitt et al., 2017*), are also not included. All of the anatomical information that could be provided by the inclusion of these taxa is lost from the early pterosaur phylogenetic analyses and ought to be corrected for.

Other studies of early pterosaur interrelationships have similar shortcomings in terms of the outgroup taxa sampling. The analysis of *Unwin (2003)*, for example, only used a single outgroup taxon in the form of the non-archosaurian archosauriform *Euparkeria capensis*. The analyses of *Kellner (2003)* had three outgroup taxa—*Ornithosuchus longidens*, *H. ischigualastensis* and *Scleromochlus taylori*. Of these, only one, *H. ischigualastensis*, is an ornithodiran. While *Scleromochlus taylori* was considered as a possible close relative of pterosaurs at the time *Kellner (2003)* was published, subsequent work on this taxon has demonstrated that it is more likely an archosauriform belonging to the clade Doswelliidae (see, *Bennett, 2020*). Finally, in the analyses of *Andres, Clark & Xu (2014)*, the chosen outgroups were the non-avemetatarsalian archosauromorphs *Euparkeria capensis* and *O. longidens*, and the putative dinosaur *H. ischigualastensis*.

This study aims to test what effect, if any, the omission of such close pterosaur relatives from analyses has had on the overall topology within Pterosauria by using a modified version of the recent dataset of *Britt et al. (2018)*. Many of the disagreements between the recent results of *Andres, Clark & Xu (2014)*, *Upchurch et al. (2015)*, *Kellner (2015)*, *Britt et al. (2018)* and *Dalla Vecchia (2014, 2019)* could be resolved through a simple addition of better and more appropriate outgroup taxa, and this is what this study attempts to do. By also incorporating new anatomical characters, taken from recent early dinosaur and archosaur studies, this study aims to better anchor the base of Pterosauria to a position within Avemetatarsalia and Ornithodira, so as to allow the 'basal' condition of pterosaurs to be better expressed in the data.

## MATERIALS AND METHODS

The dataset of *Britt et al. (2018)*, as modified by *Dalla Vecchia (2019)*, was expanded through the addition of the following taxa: Aphanosauria, Lagerpetidae, *Marasuchus lilloensis*, Silesauridae, Ornithischia, Theropoda and Sauropodomorpha.

Full details of each new operational taxonomic unit, which specimens were studied, and which other sources of anatomical information were used are given in Table 1.

**Table 1 Sources of anatomical information for taxa added to the phylogenetic analyses.**

| Operational taxonomic unit | Based upon | Specimens | Additional sources |
|---|---|---|---|
| Aphanosauria | *Teleocrator rhadinus* | NHMUK PV R6795-6 | *Nesbitt et al. (2017, 2018)* |
| | *Dongusuchus efremovi* | Multiple - PIN | *Nesbitt et al. (2017), Niedźwiedzki, Sennikov & Brusatte (2016)* |
| | *Yarasuchus deccanensis* | Multiple - ISIR | *Nesbitt et al. (2017)* |
| Lagerpetidae | *Lagerpeton chanarensis* | Multiple - PVL | *Specimens only* |
| | *D. gregorii* | TMM 31100–1306 | *Nesbitt et al. (2009a)* |
| | *D. romeri* | GR 218; DMNH EPV.29956 | *Irmis et al. (2007), Martz & Small (2019)* |
| | *D. gigas* | PVSJ 898 | *Martínez et al. (2016)* |
| | *Ixalerpeton polesinensis* | ULBRA-PVT059 | *Cabreira et al. (2016)* |
| *Marasuchus lilloensis* | *Marasuchus lilloensis* | Multiple - PVL | *Specimens only* |
| Silesauridae | *Silesaurus opolensis* | Multiple - ZPAL | *Dzik (2003)* |
| | *Kwanasaurus williamparkeri* | Multiple - DMNH | *Martz & Small (2019)* |
| | *Asilisaurus kongwe* | NHMUK R16303 | *Nesbitt et al. (2010)* |
| Ornithischia | *Hetero* | Multiple - NHMUK; SAM; BP | *Butler, Porro & Norman (2008), Norman et al. (2011), Galton (2014)* |
| | *Lesothosauru diagnosticus* | Multiple - NHMUK; BP | *Knoll (2002a, 2002b, 2002c), Porro, Witmer & Barrett (2015), Barrett et al. (2016), Baron, Norman & Barrett (2017c), Butler (2005)* |
| | *Eocursor parvus* | SAM-PK-K8025 | *Butler (2010), Butler, Smith & Norman (2007)* |
| Theropoda | *Tawa hallae* | Mutiple - GR | *Nesbitt et al. (2009b)* |
| | *Coelophysis bauri* | AMNH FR 7224 | *Specimens only* |
| | *Eodromaeus murphi* | Multiple - PVSJ | *Martínez et al. (2011)* |
| Sauropodomorpha | *Buriolestes schultzi* | CAPPA/UFSM 0035 | *Cabreira et al. (2016)* |
| | *Pampadromaeus barberenai* | ULBRA-PVT016 | *Cabreira et al. (2011)* |
| | *Saturnalia tupiniquim* | Multiple - MCP | *Langer et al. (1999), Langer (2003)* |
| | *Eoraptor lunensis* | PVSJ 512 | *Sereno, Martínez & Alcober (2013)* |
| | *Plateosaurus engelhardti* | Multiple - AMNH; SMNS | *Nesbitt (2011)* |

In addition to the new taxa, 27 new anatomical characters were incorporated into the dataset of *Britt et al. (2018)*—five were taken from the early dinosaur dataset of *Baron, Norman & Barrett (2017a, 2017b)*, which had built upon previous works (*Langer & Benton, 2006; Nesbitt, 2011*), and a further nine from the archosaur dataset of *Nesbitt et al. (2017)*. Some other characters that were added were taken from both of these studies, as they had been used in each and either entirely or partially overlapped in terms of the features that they described (chars 111–116). These characters were conflated or otherwise adjusted so as to prevent repetition or over-scoring of each feature. A further three additional characters were added based upon the range of anatomical features observed in the various taxa in the study, including a simple absent/present statement for the pteroid (char. 94) to supplement character 71 of *Britt et al. (2018)*, itself a modification from character 132 in the data of *Bennett (2013)*. In addition, a character describing the radius to humerus ratio was added (char. 110), and a character describing the shape of the distal end of the scapula (char. 95). Four more 'classic pterosaur characteristics' were accounted for
with new characters, each modified from the datasets of *Vidovic & Martill (2014)* and *Lü et al. (2009)* and were included in this study as characters 117–120. Character 99 in this analysis is a modified form of character 301 of *Baron, Norman & Barrett (2017a)*—Dorsal margin of the ilium in lateral view: 0, sinusoidal; 1, concave (saddle-shaped), pre and preacetabular and postacetabular processes upturned relative to craniocaudal centre; 2, relatively straight or convex—state 0 has been added to describe the condition seen in *Macrocnemus bassanii*, *Postosuchus kirkpatricki* and aphanosaurs. State (2) is present in theropods and ornithischians, whereas state (1) describes the condition in 'basal' pterosaurs, herrerasaurs, sauropodomorphs, silesaurids, lagerpetids and *Marasuchus lilloensis*. The full list of characters added to the data matrix is given in the Supplementary File.

Of the additional characters, 108 and 109 were treated as ordered, following *Nesbitt (2011)* and *Baron, Norman & Barrett (2017a*, *2017b)* in addition to characters 62, 74 and 91, which were also treated as ordered in the analyses of *Britt et al. (2018)* and *Dalla Vecchia (2019)*.

Trees were searched for using equal weights implementation of parsimony, using TNT 1.5-beta (*Goloboff, Farris & Nixon, 2008*), through the New technology search method. Following the protocol of *Baron, Norman & Barrett (2017a*, *2017b)* and *Nesbitt et al. (2017)*, memory was first set at its maximum of 99,999, and trees were then searched for under equal weights parsimony through a New Technology (*Goloboff, Farris & Nixon, 2008*) search, with ratchet and drift set at their default values and with 100 random additional sequences. A second search, following the protocol of *Ezcurra (2016)* was then done, in which trees were searched for using a New Technology Search (*Goloboff, Farris & Nixon, 2008*) with ratchet set to 20 iterations, with five rounds of tree fusing and 100 additional sequences. The most parsimonious trees (MPTs) produced in this second type of analysis were then subjected to a second round of TBR branch swapping, with a change probability of 33 and 100 additional sequences as the default search settings. Finally, a search was carried out using implied weights parsimony, with implied weights ($k$-values) set to 3, 5, and 10 (see *Parry, Baron & Vinther, 2017*; *Golobof, Torres & Arias, 2018*).

This manuscript and the nomenclatural acts it contains was registered with ZooBank and the manuscript assigned the following LSID: urn:lsid:zoobank.org:pub:BE350658-1D5C-456B-B129-FFDE827E7DDF.

## RESULTS

An initial analysis was run that excluded the additional 27 characters that were to be added to the dataset of *Britt et al. (2018)*. This was done using equal weights and a simple New technology search. This analysis was carried out to test the effect of an expanded set of outgroup taxa alone, without the effect of added characters. The analysis produced 31 MPTs each of length 305 steps (Fig. 2). In spite of the lack of additional characters that could help to resolve the relationships within Ornithodira and Avemetatarsalia, this analysis still recovered a monophyletic Pterosauria and generated fairly good resolution within this clade. The resolution among outgroup taxa is poor, with most outgroup taxa forming a polytomy outside of Pterosauria. Within Pterosauria there exists a

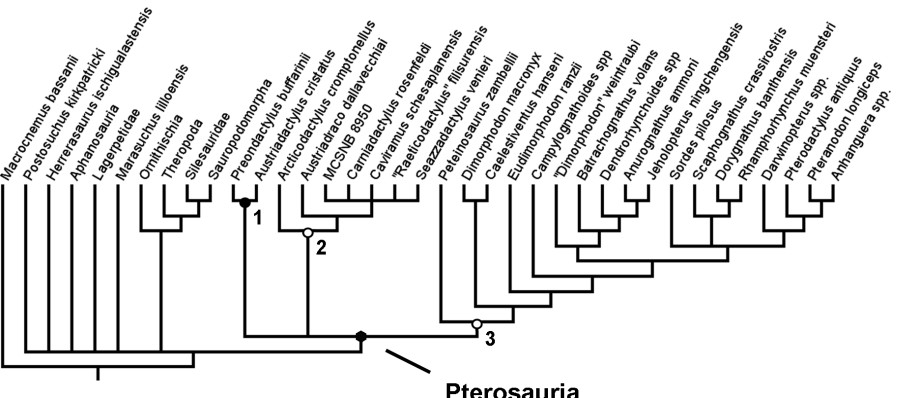

**Figure 2 Strict consensus rule tree produced in the initial analysis that did not utilise any new anatomical characters.** Nodes: 1, Preondactylia; 2, Caviramidae (*clade novo*); 3, Zambellisauria (*clade novo*).

'basal' trichotomy. *Austriadactylus cristatus* and *Preondactylus bufarinii* are recovered in a sister taxon relationship; a second 'basal' clade comprises *Arcticodactylus cromptonellus*, *Austriadraco dallavecchiai*, *Seazzadactylus venieri*, *Carniadactylus rosenfeldi*, '*Raeticodactylus*' *filisurensis*, *Caviramus schesaplanensis* and unnamed specimen MCSNB 8950; the third of the 'basal' clades in the trichotomy contains *Peteinosaurus zambelli*, Dimorphodontidae and Lonchognatha. This result is more similar to the results of the analysis carried out by *Dalla Vecchia (2019)* than those of *Britt et al. (2018)*, though this analysis has poorer resolution at the base of the pterosaur tree. The addition of new outgroups has, without the addition of new characters, generated more uncertainty about the fundamental interrelationships between the earliest diverging pterosaur groups. Furthermore, the resolution between taxa in the second clade produced in this analysis—the one containing *Arcticodactylus cromptonellus*, *Austriadraco dallavecchiai*, *Seazzadactylus venieri*, *Carniadactylus rosenfeldi*, '*Raeticodactylus*' *filisurensis*, *Caviramus schesaplanensis* and specimen MCSNB 8950—is poorer with the addition of the new outgroups. In this analysis *Arcticodactylus cromptonellus* and *Austriadraco dallavecchiai* from a grade leading to a polytomy containing all other taxa in this group. In the results of Dalla Vecchia, on the other hand, found *Seazzadactylus venieri* and *Carniadactylus rosenfeldi* to also for part of this grade leading to a smaller polytomy of *Raeticodactylus*' *filisurensis*, *Caviramus schesaplanensis* and specimen MCSNB 8950. Within the other clades the recovered topology is the same as in the analyses of *Dalla Vecchia (2019)* (Fig. 1B). The addition of new outgroup taxa alone did not result in the recovery of a monophyletic Austriadraconidae or Eopterosauria, as in other previous studies (*Andres, Clark & Xu, 2014*; *Britt et al., 2018*) (Figs. 1A and 1C). By adding in new characters that better resolve the relationships within Ornithodira and the character optimisation at the base of Pterosauria, this uncertainty at the base of the pterosaur tree was resolved and a different topology within certain constituent pterosaurian clades was recovered (Fig. 3).

In this first full analysis that included both the added taxa and added characters, and using equal weights and a simple New technology search, two MPTs were recovered, each

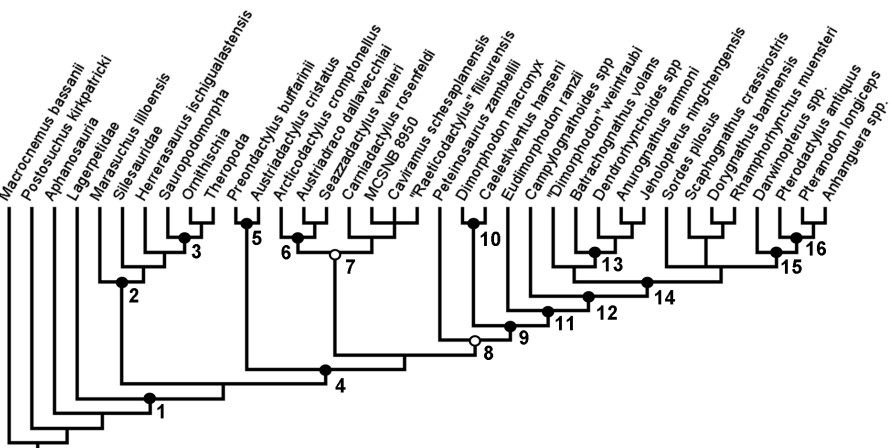

**Figure 3 Strict consensus rule tree produced in full analysis one, using equal weights parsimony.**
Nodes: 1, Dinosauromorpha; 2, Dinosauriformes; 3, Dinosauria; 4, Pterosauria; 5, Preondactylia; 6, Austriadraconidae; 7, Caviramidae (*clade novo*); 8, Zambellisauria (*clade novo*); 9, Macronychoptera; 10, Dimorphodontidae; 11, Lonchognatha; 12, Novialoidea; 13, Anurognathidae; 14, Caelidracones; 15, Monofenestrata; 16, Pterodactyloidea.            

of length 390 steps. In the strict consensus rule tree produced from the two MPTs recovered in the analysis, a monophyletic Pterosauria was found (Fig. 3). This clade contains all the taxa analysed in this analysis that are traditionally considered to be pterosaurs, and together this clade forms a sister-taxon to a clade containing almost all of the newly added avemetatarsalian taxa, except for Aphanosauria and Lagerpetidae. Dinosauria is recovered, as is Dracohors and Dinosauriformes. Lagerpetidae is recovered without the clade containing Dinosauriformes and Pterosauria. *H. ischigualastensis*, which was the only non-pterosaurian ornithodiran outgroup included in the analyses of *Britt et al. (2018)* and *Dalla Vecchia (2019)*, is found nested within Dracohors, in a position closer to Dinosauria than to Silesauridae, as is more consistent with some recent analyses of early dinosaurs (*Baron & Williams, 2018*).

In this full analysis, with all of the additional taxa and characters being active, the base of the pterosaurian clade no longer contained a trichotomy. Instead, *Austriadactylus cristatus* and *Preondactylus bufarinii* are recovered as sister-taxa, forming their own small monophyletic group at the base of the pterosaur tree, falling outside of the clade that contains all other pterosaurs sensu *Dalla Vecchia (2019)*. This clade—named Preondactylia by *Andres, Clark & Xu (2014)*—has also been found in a number of other studies (*Upchurch et al., 2015*; *Britt et al., 2018*; *Dalla Vecchia, 2019*). Preondactylia forms the sister taxon to a clade containing two distinct monophyletic groups: one group contains *Arcticodactylus cromptonellus*, *Austriadraco dallavecchiai*, *Seazzadactylus venieri*, *Carniadactylus rosenfeldi*, 'Raeticodactylus' *filisurensis*, *Caviramus schesaplanensis* and specimen MCSNB 8950; the other contains *Peteinosaurus zambelli* and all other pterosaurs. This too largely agrees with the results obtained by *Dalla Vecchia (2019)*—however, the topology within the first of the two clades differs. As discussed above, in the results presented by *Dalla Vecchia (2019)*, *Arcticodactylus cromptonellus*, *Austriadraco dallavecchiai*, *Seazzadactylus venieri* and *Carniadactylus rosenfeldi* formed a grade leading

to a clade containing 'Raeticodactylus' filisurensis, Caviramus schesaplanensis and specimen MCSNB 8950. This expanded analysis did not find such a topology within this clade. Instead, the results recover Arcticodactylus cromptonellus, Austriadraco dallavecchiai and Seazzadactylus venieri in their own clade, which is sister-taxon to a clade containing the others. This first clade is akin to Austriadraconidae, as named by Kellner (2015). Austriadraconidae, in this form, is not supported in the results presented by Dalla Vecchia, but was recovered, albeit in a different position by Britt et al. (2018). The placement and composition of Austriadraconidae in the results of this analysis are novel and appear to be the result of the combination of wider outgroup sampling and anatomical character choice. Within other major pterosaurian sub-clades, such as Macronychoptera, Dimorphodontidae, Anurognathidae, and Pterodactyloidea, the topology recovered in this analysis agrees with the topology recovered in the analysis by Dalla Vecchia (2019). Moreover, this analysis found no support for the placement of Arcticodactylus cromptonellus and Austriadraco dallavecchiai in their own exclusive clade placed as sister-taxon to all other pterosaurs, as had been found by Britt et al. (2018), sensu Dalla Vecchia (2019).

## Further comparisons with previous studies

While the placement of Preodactylia as the earliest diverging of the pterosaur subclades agrees with the analysis by Dalla Vecchia (2019), the result differs from Britt et al. (2018). Whereas in the taxon-reduced analyses of Britt et al. (2018), a clade containing Austriadraco dallavecchiai and Arcticodactylus cromptonellus—termed Austriadraconidae by Kellner (2015)—was found as the earliest diverging pterosaur clade, the results of Dalla Vecchia (2019) and those of the first analysis of this study do not provide support for such a position, but rather place the austriadraconids in a more deeply nested position. This result also differs substantially from that found by Codorniú et al. (2016) who did not find a sister-taxon relationship between Austriadactylus cristatus and Preondactylus bufarinii at all.

As stated above, the results of this first analysis support a monophyletic Austriadraconidae, sensu Kellner (2015), but, for the first time, also places Seazzadactylus venieri within it (Fig. 2). In this analysis, the austriadraconids form a sister-taxon to a clade containing Carniadactylus rosenfeldi + 'Raeticodactylus' filisurensis, Caviramus schesaplanensis and unnamed specimen MCSNB 8950. Dalla Vecchia (2019) also recovered all of these taxa together into a monophyletic group, but his analysis did not recover the same interrelationships between them; Arcticodactylus cromptonellus, Austriadraco dallavecchiai and Seazzadactylus venieri form a grade leading into the clade containing Carniadactylus rosenfeldi and a trichotomy of 'Raeticodactylus' filisurensis, Caviramus schesaplanensis and MCSNB 8950 in the analysis of Dalla Vecchia (2019) (see, Fig. 1A).

No close relationship is found between this group of pterosaurs and Eudimorphodon ranzii, contra the findings of Andres, Clark & Xu (2014), Upchurch et al. (2015) and Britt et al. (2018). Similarly, no support is found for the clade Eudimorphodontidae.

Instead, *Eudimorphodon ranzii* is found within Macronychoptera and Lonchognatha, sensu *Dalla Vecchia (2019)*.

Also consistent with the analysis of *Britt et al. (2018)* and of *Dalla Vecchia (2019)*, *D. macronyx* and *Caelestiventus hanseni* are found to be sister-taxa, forming their own clade Dimorphodontidae. This stands in contrast to the results of *Codorniú et al. (2016)*, who recovered *Peteinosaurus zambelli* as the sister-taxon to *D. macronyx*.

Both the analyses by *Britt et al. (2018)* and *Dalla Vecchia (2019)* found a Macronychoptera containing Dimorphodontidae and Lonchognatha, sensu *Unwin (2003)*, and with *Peteinosaurus zambelli* forming the sister-taxon to Macronychoptera. This again differed from the results of *Codorniú et al. (2016)* who found a different position for *Peteinosaurus zambelli* (see above). However, despite the agreement between *Britt et al. (2018)* and *Dalla Vecchia (2019)* on the subgroups comprising Macronychoptera, the constituent taxa of Lonchognatha differed between these two analyses, with *Britt et al. (2018)* finding taxa such as '*Raeticodactylus*' *filisurensis* and *Caviramus schesaplanensis* to be members of this more 'derived' clade (see also, *Kellner, 2003*). This study has not found such a construction of Lonchognatha, and instead has found a more reduced clade, sensu *Dalla Vecchia (2019)*. Further, within Lonchognatha is a clade containing Campylognathoides and all other pterosaurs analysed in this study—that is, a monophyletic Novialoidea sensu *Kellner (2003)*—and this result largely agrees with the analyses of *Kellner (2003)*, *Andres & Myers (2013)*, *Andres, Clark & Xu (2014)*, *Upchurch et al. (2015)*, *Britt et al. (2018)* and *Dalla Vecchia (2019)*.

Also consistent with each of the above studies is the recovery of a monophyletic Caelidracones. However, a sister-taxon relationship between Anurognathidae and Pterodactyloidea, as recovered by *Andres & Myers (2013)* is not supported in this analysis. As in the analyses by *Britt et al. (2018)* and *Dalla Vecchia (2019)*, Caelidracones contains two clades: one containing a trichotomy of *Sordes pilosus*, 'ramphorynchids', and Monofenestrata, and the other containing taxa that could be loosely termed 'anurognathid types'. Again, a contrast can be drawn with the results of this analysis and the results of the analysis by *Codorniú et al. (2016)*, who placed Anurognathidae in a much more stem-ward position within Pterosauria.

Anurognathidae is also recovered in this analysis, but the interrelationships between the taxa in this clade differ from the analyses of *Britt et al. (2018)* and *Dalla Vecchia (2019)* in that *Jeholopterus* and *Anurognathus* form sister taxa, with *Dendrorynchoides* and *Batrachognathus* forming successive sister-taxa. The taxon provisionally named '*Dimorphodon*' *weintraubi* then forms the sister taxon to Anurognathidae, and this is consistent with the results of *Dalla Vecchia (2019)*. These 'anurognathid types' together form the sister-taxon to the grouping of *Sordes pilosus*, the 'ramphorynchids' and monofenestratans, which is, as yet, also unnamed (see *Dalla Vecchia, 2019*).

One interesting aspect of this analysis is that it would appear, when applying current definitions, that Pterosauria falls within Dinosauromorpha sensu *Benton (1985)* and not as its sister-taxon, sensu *Gauthier (1986)* and *Nesbitt et al. (2017)*. Because Lagerpetidae falls outside of the clade containing Pterosauria and the grade leading into

dinosaurs in this analysis, under the current definition of Dinosauromorpha—the last common ancestor of *Lagerpeton chanarensis*, *Marasuchus lilloensis*, Dinosauria and all its descendants (*Benton, 1985*)—Pterosauria and, by definition, Ornithodira, would fall within Dinosauromorpha. Ornithodira would comprise Pterosauria and Dinosauriformes, rather than Pterosauria and Dinosauromorpha, as suggested in previous analyses (*Langer & Benton, 2006*; *Nesbitt, 2011*; *Baron, Norman & Barrett, 2017a*; *Nesbitt et al., 2017*). However, this position is only relatively weakly supported for now, and such interaltionships within the higher Ornithodira was not the focus of this analysis. Subsequent studies may recover more traditional topologies within Avemetatarsalia and composition of Ornithodira, so this study refrains from revising any definitions based upon this result alone. Within Dinosauriformes are *Marasuchus lilloensis* and Dracohors, sensu *Cau (2018)*. Silesaurids form the sister-taxon to dinosaurs + herrerasaurs, sensu *Baron & Williams (2018)*. Dinosauria in this analysis contains Ornithoscelida (see *Baron, Norman & Barrett, 2017a*) and Sauropodomorpha (see *Baron, Norman & Barrett, 2017b*).

The following is a list some of the notable nodes recovered from the base of the pterosaur tree and including Pterosauria, with all synapomorphies listed for each (Node numbers refer to numbering in Fig. 3):

**Node 4. Pterosauria**, sensu *Kellner (2003)* ((*Preondactylus bufarinii* + *Austriadactylus cristatus*)/Preondactylia + (Caviramidae + Zambellisauria))

Definition: Node-based—the most recent common ancestor of the Anurognathidae, *Preondactylus bufarinii* and *Quetzalcoatlus northropi* and all their descendants (*Kellner, 2003*).

Character support: 7 (1–>3), 9 (0–>1), 16 (1–>2), 22 (0–>1), 39 (0–>1), 40 (0–>1), 58 (0–>1), 60 (0–>1), 70 (0–>1), 74 (0–>1), 75 (0–>1), 91 (0–>1), 94 (0–>1), 117 (0–>1), 120 (2–>0).

Remarks: The definition given by *Kellner (2003)* is sufficient to contain all taxa found in this analysis to be contained within the pterosaur group. As *Preondactylus bufarinii*, together with *Austriadactylus cristatus*, forms the earliest diverging clade within Pterosauria—a monophyletic Preondactylia—this definition for Pterosauria encompasses the same set of taxa as Pterosauromorpha, sensu *Padian (1997)*. Pterosauria would, in this hypothesis, take precedence over Pterosauromorpha. However, it should be worth noting that, in results that do not find Preondactylia to be the earliest diverging pterosaur clade (*Britt et al., 2018*), Pterosauria would encompass fewer taxa than Pterosauromorpha, as certain clades would fall outside of the taxa encompassed by the definition for Pterosauria given by *Kellner (2003)*. In the example of *Britt et al. (2018)*, Austriadraconidae would not be within Pterosauria but rather would be non-pterosaurian pterosauromorphs, under such a regime (see below).

**Node 5. Preondactylia**, sensu *Andres, Clark & Xu (2014)* (*Preondactylus bufarinii* + *Austriadactylus cristatus*)

Definition: The least inclusive clade that includes *Preondactylus bufarinii* and *Austriadactylus cristatus*.

Character support: 11 (0–>1), 48 (0–>1), 49 (0–>1).

Remarks: Often found to be the earliest diverging members of Pterosauria (*Dalla Vecchia, 2019*), or early diverging members of 'Eopterosauria' (*Andres, Clark & Xu, 2014*; *Upchurch et al., 2015*), this small clade is fairly consistently recovered among early pterosaur cladistic studies, with the notable exception of *Codorniú et al. (2016)*.

**Unnamed clade** (Caviramidae + Zambellisauria)

Character support: 45 (1–>0), 47 (1–>0), 77 (0–>1), 95 (0–>1).

Remarks: This clade contains all pterosaur taxa except for the clade containing *Preondactylus bufarinii* + *Austriadactylus cristatus* that is, all pterosaurs more 'derived' than Preondactylia. This clade has consistently been found among studies (*Codorniú et al., 2016*; *Britt et al., 2018*; *Dalla Vecchia, 2019*), although the placement of certain taxa within this clade varies, for example in the results of *Britt et al. (2018)*, this clade does not contain *Arcticodactylus cromptonellus* and *Austriadraco dallavecchiai* (see Fig. 1C). It may prove necessary in the future to erect a stem-based clade to contain all taxa more closely related to, as an example, *Q. northropi* than to *Preondactylus bufarinii*.

**Node 7: Caviramidae** (new clade)

Definition: Node-based—the least inclusive clade that includes *Arcticodactylus cromptonellus* and *Caviramus schesaplanensis* (new).

Etymology: For *Caviramus schesaplanensis*—one of two anchoring member taxa used in the node-based definition for the clade, as outlined above.

Character support: 3 (0–>1), a skull that is curved down caudally; 12 (0–>1), a jugal process of the maxilla that is subtrapezoidal, tapering to a point only distally, and a proximal part with parallel dorsal and ventral margins; 43 (0–>1), a dentition that has tri- to quinticuspid tooth crowns; 81 (0–>1), a wing phalanx two that is as long as the ulna.

Remarks: This clade has been recovered in the recent analysis of *Dalla Vecchia (2019)* and is also supported in the first analysis of this study by a number of shared anatomical character states, or synapomorphies (see above). The topology within the clade varies between the analyses of *Dalla Vecchia (2019)* and this study, but both find the same set of taxa and specimens to fall within it. This early diverging subgroup contains within it the austriadraconids of *Kellner (2015)* and a handful of other Triassic taxa and specimens, including the as yet unnamed MCSNB 8950, which has previously been the source of phylogenetic uncertainty in other studies (see *Britt et al., 2018*). This clade is the least consistently supported among the various analyses of this study and this is discussed further below. However, it is worth noting that the clade was found in both the analysis that used only new outgroup taxa but no new characters and the analysis that used both.

The internal topology differed between these two analyses, as did the position of the clade within Pterosauria, but the constituent taxa was consistent, and consistent with the results of *Dalla Vecchia (2019)*.

**Node 6. Austriadraconidea**, sensu *Kellner (2015)* (*Arcticodactylus cromptonellus* + *Austriadraco dallavecchiai* + *Seazzadactylus venieri*)

Definition: The least inclusive clade that includes *Arcticodactylus cromptonellus* and *Austriadraco dallavecchiai*, sensu *Kellner (2015)*.

Character support: 103 (1–>0), 120 (0–>1).

Remarks: This clade was named by *Kellner (2015)*, but was not supported in a number of other recent analyses of early pterosaurs (*Andres, Clark & Xu, 2014*; *Upchurch et al., 2015*). However, this close relationship was found in the more recent analyses of *Britt et al. (2018)*, who recovered the clade as the earliest diverging within Pterosauria, and then *Dalla Vecchia (2019)*, who, like this study, found the clade to be slightly more 'derived'. Unlike either of the aforementioned studies, this study has found that the clade also contains *Seazzadactylus venieri*; as in the analysis of *Dalla Vecchia (2019)* this grouping of taxa fall within a larger clade of early-diverging pterosaur taxa (here named as Caviramidae, clade nov.).

**Unnamed clade** (*Carniadactylus rosenfeldi* + 'Raeticodactylus' filisurensis, *Caviramus schesaplanensis* + MCSNB 8950)

Character support: 11 (0–>2), 21 (0–>1), 65 (0–>1), 76, (0–>1), 80 (0–>1).

Remarks: This clade was also recovered in the analysis by *Dalla Vecchia (2019)* and contained the same taxa and the same internal topology.

**Node 8. Zambellisauria** (new clade) (*Peteinosaurus zambelli* + Macronychoptera)

Definition: Node-based—the least inclusive clade that includes *Peteinosaurus zambelli*, *D. macronyx*, *Pterodactylus antiquus* and *Q. northropi* (new).

Etymology: The clade name honours Rocco Zambelli, curator of the Bergamo natural history museum, for whom *Peteinosaurus zambelli* was named; *Peteinosaurus zambelli* being one of the two taxa chosen as an anchor in this cladistic definition. *Zambelli (1973)* also named *Eudimorphodon ranzii*, a well-known and important early pterosaur and putative member of Zambellisauria (see *Andres, Clark & Xu, 2014*; *Britt et al., 2018*; *Dalla Vecchia, 2019*).

Character support: 56 (0–>1), more than three sacral vertebrae; 59 (0–>1), filiform processes of the caudal zygapophyses present in caudal vertebrae.

Remarks: This clade is consistently recovered by most modern analyses (*Britt et al., 2018*, *Dalla Vecchia, 2019*) and in all of the analyses in this study, regardless of the optimality criteria used in searching for trees in the analyses, and both with and without the inclusion

of the new anatomical characters. Such a clade was not found in the analysis of *Andres, Clark & Xu (2014)*, who instead placed *Peteinosaurus zambelli* as sister-taxon to Eudimorphodontoidae, within a monophyletic Eopterosauria—a hypothesis that has fallen out favour in more recent studies and is not supported in any of the analyses carried out in this study.

**Node 9. Macronychoptera** (Dimorphodontidae + Lonchognatha)

Character support: 63 (1->2), 64 (0->1), 65 (0->1)

**Node 10. Dimorphodontidae** (*D. macronyx* + *Caelestiventus hanseni*)

Character support: 7 (3–>2), 8 (0–>1), 10 (0–>1), 12 (0–>2), 20 (0–>1), 26 (0–>1), 35 (0–>1), 41 (0–>1), 48 (0–>1), 52 (0–>1), 96 (1–>0), 118 (1–>2).

Remarks: As in the analyses of *Britt et al. (2018)* and *Dalla Vecchia (2019)*, this study finds a close relationship between *D. macronyx* and *Caelestiventus hanseni*

**Node 11. Lonchognatha**

Character support: 70 (1–>2), 80 (0–>2), 88 (0–>1)

Remarks: The composition of Lonchognatha and its possible sister-taxon relationship with Dimorphodontidae has been fairly consistently recovered in recent analyses (*Britt et al., 2018*; *Dalla Vecchia, 2019*), although other studies have placed Lonchognatha within Novialoidae (*Andres, Clark & Xu, 2014*). This second hypothesis is not recovered in all but one of the analyses of this study. That is to say, in only one analysis does this study find *Campylognathoides* to be 'less derived' than *Eudimorphodon* species (see below).

In the second full analysis, utilising both the new taxa and new anatomical characters, TBR branch swapping after a differently configured New Technology search produced two MPTs, each also of length 390 steps. However, in this analysis, the resolution in this tree was greatly reduced (Fig. 4A). While Aphanosauria and Lagerpetidae were still found outside of Ornithodira, and the monophyly of and interrelationships within Dinosauriformes remained consistent with previous analyses, in this second full analysis, the interrelationships between the groups within Pteorsauria was not clearly resolved—although monophyletic Austriadraconidae and Novialoidea were both found again. The large polytomy at the base of Pterosauria is more reminiscent of the full-taxon-sample analysis carried out using TNT by *Britt et al. (2018)*. As in *Britt et al. (2018)*, the specimen MCSNB 8950 was removed as a wildcard taxon and the analysis re-run. In this reduced analysis, a single tree was recovered (Fig. 4B). In this tree, Austriadraconidae is once again recovered as monophyletic, as in the first analysis of this study and the second full-taxon-range analysis with TBR. However, in the reduced second analysis, but unlike in the full second analysis, Austriadraconidae contains *Seazzadactylus venieri*. This is similar to the result obtained in the first analysis, although the position of Austriadraconidae is different in both the reduced and full second analyses (compare Figs. 3 and 4B).
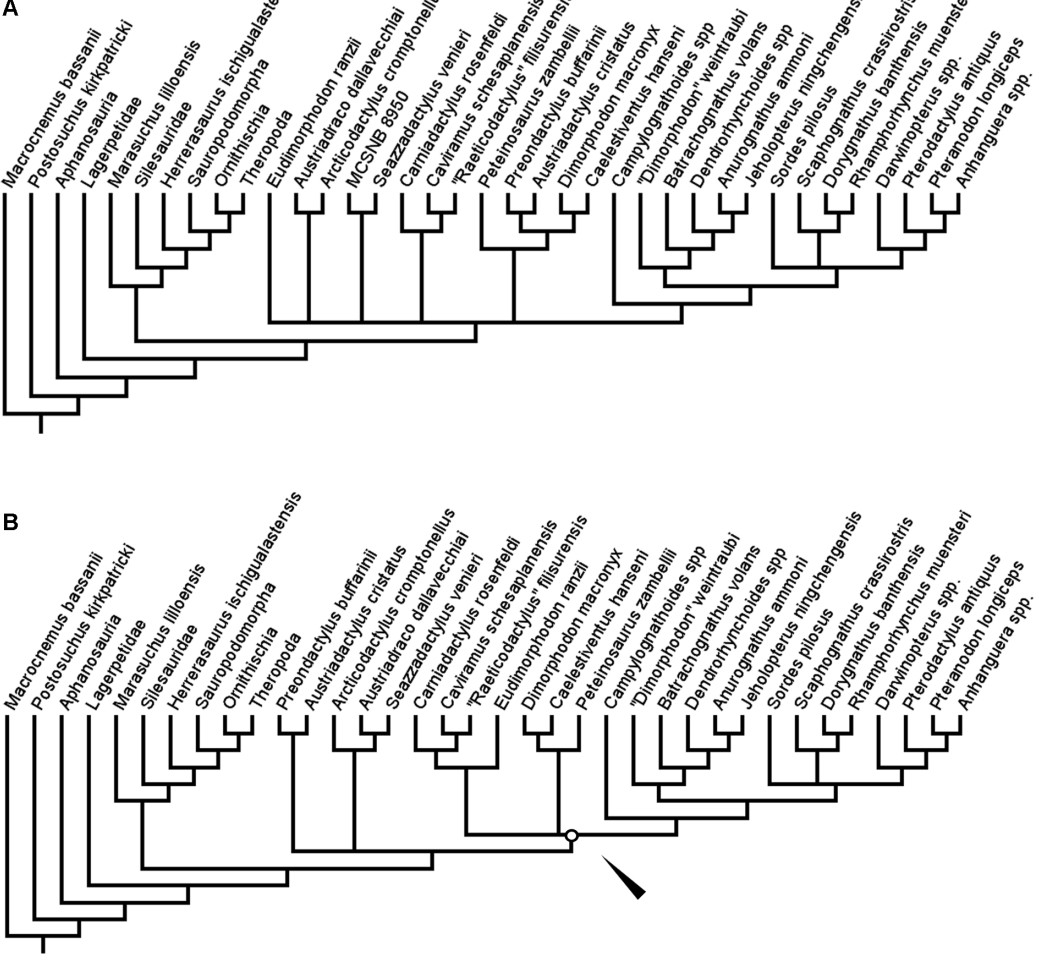

**Figure 4 Strict consensus (A) and reduced strict consensus (B) trees produced when following the analysis protocol of Ezcurra (2014) and using equal weights parsimony.** White circle and arrow indicate the position of Zambellisauria in this analysis.   

   Finally, in the implied weights analyses, the topology within Pterosauria differed from the first and second analyses (Figs. 5A and 5B). Only one tree was produced in each of three analyses, with lengths of 37.03690, 27.65440, and 17.04634 for *k* = 3, 5 and 10 respectively. In the first two of these analyses (*k* = 3 and 5), the earliest diverging clade of pterosaurs are the Austriadraconidae—however, this clade was not found to contain *Seazzadactylus venieri* in this analysis. The clade made up of *Austriadactylus cristatus* and *Preondactylus buffarini*, named above as Preondactylidae, then forms the sister-taxon of the clade of *Peteinosaurus zambelli* + Macronychoptera, named above as Zambellisauria. In searches with *k* at or above 10 however, the structure of the tree once again changes, with the preondactylids once again falling out as the most stem-ward of the clades in Pterosauria and, as in some of the previous analyses of this study, Austriadraconidae was found to contain *Seazzadactylus venieri* (Fig. 5B). Caviramidae was not supported in any of these analyses, whereas Zambellisauria was consistently recovered in each. Likewise, clades such as Dimorphodontidae, Lonchognatha, Novialoidea, Caelidracones,

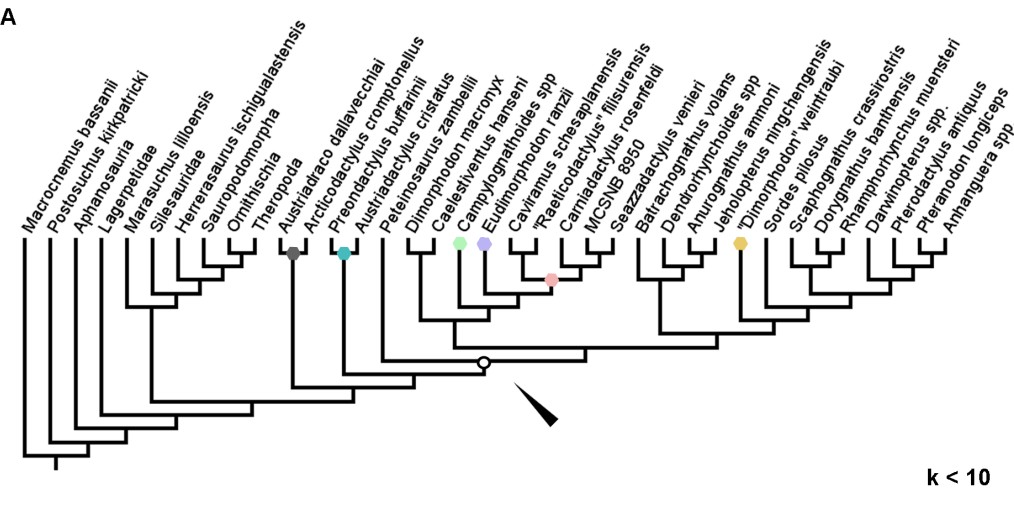

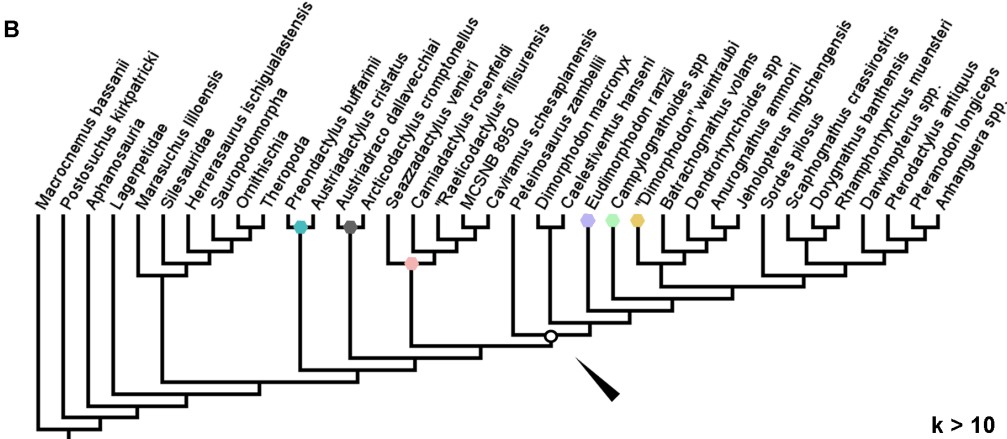

**Figure 5 Trees produced using implied weights implementation of parsimony for *k* values < 10 (A) and *k* values = 10 more (B).** Coloured nodes and tips added to highlight the taxa whose positions changed substantially between the implied weights parsimony analyses. White circles and arrows indicate the position of Zambellisauria in each analysis.

Monofenestrata, Pterodactyloidea, and Anurognathidae were all found to be largely consistent through the various implied weights parsimony analyses. The taxon referred to as '*Dimorphodon*' *weintraubi* was recovered closer to *Sordes pilosus* and the monofenestratans than to Anurognathidae in the lower weighted implied weights analyses, but closer to Anurognathidae in analyses with a *k* value = 10 or more. In all the implied weights analyses, the 'ramphorynchids' we found to be more closely related to Monofenestrata than to *Sordes pilosus*, which was recovered as the sister-taxon to the clade of Monofenestrata + the 'ramphorynchids' (Figs. 5A and 5B).

The largest difference in the arrangement of taxa between the implied weights analyses for *k* < 10 and for *k* = 10 or more is the composition of Zambellisauria. In all the analyses Zambellisauria is recovered as monophyletic, but in the lower *k* value analyses, contains, inter alia, *Seazzadactylus venieri*, *Carniadactylus rosenfeldi*, '*Raeticodactylus*' *filisurensis*, *Caviramus schesaplanensis* and the specimen MCSNB 8950, whereas in

analyses with $k = 10$ or more, these five operational taxonomic unites were recovered outside of Zambellisauria (see Figs. 5A and 5B). These differences in topology between the results of the various implied weights parsimony analyses suggests that certain anatomical characters in the dataset are broadly distributed across the taxa whilst also being important for uniting certain clades—as the implied weighting factor is increased, the weight of characters that could support the monophyly of clades such as Caviramidae (see above), or support a 'basal' position for Austriadraconidae, are being reduced by the search programme because these characters appear to be more homoplastic. However, these effects may be reduced by the further addition of anatomical characters as the dataset is expanded upon in subsequent studies.

## DISCUSSION

It is clear from the variability in the results of the analyses in this study alone that the phylogenetic position of certain early pterosaur clades is still highly unstable, even with better taxon and character sampling. When analysed using certain methods, the addition of more avemetatarsalian taxa and new characters provided further support for certain clades and helped to revolve interrelationships between genera within some subclades. In addition, this expanded analysis changed the composition of some recognised clades, from example Austriadraconidae, which was, in analysis one, found to also include the recently described *Seazzadactylus venieri* for the first time. Choice of taxa and anatomical characters clearly has had some effect on the interrelationships of the ingroup pterosaur taxa, as has also been demonstrated to be the case in early studies of dinosaur (*Müller & Dias-da-Silva, 2019*). This study has taken steps to address the under-sampling from closely related ornithdiran clades in previous pterosaur studies, but much work needs to be done to further broaden the datasets used in phylogenetic analyses, in terms of both the operational taxa and anatomical characters and character states.

What this study has also demonstrated is how using different approaches to phylogenetic analysis can produce substantially different results when it comes to the interrelationships within Pterosauria. The earliest diverging clade of pterosaurs has been found to be either the Preondactylia (in analysis one and in implied weights analyses for $k = 10$ or more), or Austriadraconidae (when using TBR branch swapping or implied weights with $k < 10$).

However, in spite of this uncertainty, some clades have been consistently recovered throughout the various analyses of this study, and many too have been supported in previous analyses (*Andres, Clark & Xu, 2014*; *Britt et al., 2018*; *Dalla Vecchia, 2019*). This would suggest that the evidence is increasingly supporting the validity of such monophyletic subsets within Pterosauria, and these should, for the sake of stability and clarity in future research, be defined and, if not already so, named. This study has erected two clades for these purposes. Zambellisauria is erected to contain all pterosaur taxa descended from the most recent common ancestor of *Peteinosaurus zambellii* and the various members of Macronychoptera. This clade is now consistently recovered in most phylogenetic analyses and is strongly supported in the results of these analyses. Should *Peteinosaurus zambellii* be recovered in a much more stem-ward position in the future, for

example if future analyses resurrect Eopterosauria, with *Peteinosaurus zambellii* contained within it, the definition for Zambelliasauria would then encompass the same set of taxa as Pterosauria and, as a result, would become obsolete. However, in the emerging consensus on early pterosaur relationships in this part of the tree, the clade Zambellisauria remains well-supported and distinct for now.

Under the definition given for Caviramidae, as also given above, in scenarios in which a distinct monophyletic group is found to be more 'derived' than Preondactylia but outside of the clade containing all other, more 'derived' pterosaurs (see Fig. 3), the name would be distinct and valid. However, in scenarios such as presented in the results of the second and third analyses, or in other studies (*Britt et al., 2018*) the clade Caviramidae would contain the same taxa as Pterosauria and therefore become obsolete. However, the validity of Caviramidae is not dependent on the position taxa such as *Peteinosaurus zambellii* and *Eudimorphodon ranzii*, which have a tendency to 'bounce around' the tree, so long as the Autriadraconids are not found to be either more or less 'derived' than the small clade containing *Carniadactylus rosenfeldi*, 'Raeticodactylus' *filisurensis*, and *Caviramus schesaplanensi*. It is only in hypotheses in which Austriadraconids fall as the earliest diverging members of Pterosauria (e.g. analysis two of this study; *Britt et al., 2018*), or are paraphyletic (*Codorniú et al., 2016*), that Caviramidae would be invalid as a distinct clade.

It would also appear from the results of these analyses, and other recent works, that the higher-level interrelationships between pterosaur taxa are becoming more stabilised. Clades such as Dimorphodontidae, Lonchognatha, Novialoidea, Caelidracones, Monofenestrata, Pterodactyloidea, and Anurognathidae are all consistently found in these analyses, and only the composition of each varies a little between them. In particular *Eudimorphodon ranzii* is a particularly difficult taxon in terms of its position, having been recovered in a range of 'derived' and more 'basal' positions within the tree, in the both the analyses of this study and previous recent studies. With the addition of more taxa and more characters, and as more phylogenetic analysis techniques are turned on the question of pterosaur systematics, such problems of placing difficult to classify taxa may yet be resolved.

## CONCLUSIONS

Pterosaur interrelationships have been shown to vary between analyses, with the fundamental interrelationships that are recovered being dependant upon the method of analysis, the character choice and taxon choice. The addition of more appropriate avemetatarsalian outgroup taxa to the early pterosaur dataset of *Britt et al. (2018)* made a difference in the overall topologies recovered within the various pterosaur clades, and to the fundamental structure of the pterosaur tree. However, a more dramatic change in result could be achieved through the use of different phylogenetic analysis techniques, such and implied weights parsimony. While some pterosaur clades have proven to be stable throughout the various analyses, others have not, particularly those that fall most stem-ward on the tree. More needs to be done to resolve this issue, but wider character and taxon sampling in the future would be an important first step. Additional, utilisation of a

wider range of phylogenetic analysis techniques should be adopted to test the strength of hypotheses of early pterosaur interrelationships as more taxa and character states are added.

### Funding
The authors received no funding for this work.

### Competing Interests
The authors declare that they have no competing interests.

### Author Contributions
- Matthew G. Baron conceived and designed the experiments, performed the experiments, analysed the data, prepared figures and/or tables, authored or reviewed drafts of the paper, and approved the final draft.

### Data Availability
The newly incorporated anatomical characters, character modifications and character scores are available in the Supplemental Files.

The specimens are as follows:

*Teleocrator rhadinus*—NHMUK PV R6795-6

*Dongusuchus efremovi*—PIN 952/15-1-/15-6; PIN 952/84-1-/84-6

*Yarasuchus deccanensis*—ISI R 334/37; ISI R 334/56; ISI R 334/63; ISI R 334/9; ISI R 334/36

*Lagerpeton chanarensis*—PVL 4619; PVL 4625

*D. gregorii*—TMM 31100–1306

*D. romeri*—GR 218; DMNH EPV.29956

*D. gigas*—PVSJ 898

*Ixalerpeton polesinensis*—ULBRA-PVT059

*Marasuchus lilloensis*—PVL 3871

*Silesaurus opolensis*—ZPAL Ab III/361; ZPAL Ab III/361/35; ZPAL Ab III/361/36

*Kwanasaurus williamparkeri*—DMNH EPV.65879, A-H; DMNH EPV.63136; DMNH EPV.59302; DMNH EPV.67956; DMNH EPV.125924 (A-E)

*Asilisaurus kongwe*—NHMUK R16303

*Heterodontosaurus*—SAM-PK-K337; SAM-PK-K10487

*Lesothosauru diagnosticus*—NHMUK PV RU B17; NHMUK PV RU B23; NHMUK PV R11956; SAM-PK-K1105; SAM-PKK1106; BP/1/4885

*Eocursor parvus*—SAM-PK-K8025

*Tawa hallae*—GR 241

*Coelophysis bauri*—AMNH FR 7224

*Eodromaeus murphy*—PVSJ 560

*Buriolestes schultzi*—CAPPA/UFSM 0035

*Pampadromaeus barberenai*—ULBRA-PVT016

*Saturnalia tupiniquim*—MCP 3844-PV; MCP 3845-PV; MCP 3846-PV

*Eoraptor lunensis*—PVSJ 512

*Plateosaurus engelhardti*—SMNH 12949; SMNS 13200; AMNH 6810

## New Species Registration

The following information was supplied regarding the registration of a newly described species:

Publication LSID: urn:lsid:zoobank.org:pub:BE350658-1D5C-456B-B129-FFDE827E7DDF.

## Supplemental Information

Supplemental information for this article can be found online at http://dx.doi.org/10.7717/peerj.9604#supplemental-information.

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
