# Peer review of "Testing pterosaur ingroup relationships through broader sampling of avemetatarsalian taxa and characters and a range of phylogenetic analysis techniques"

_PeerJ, doi:10.7717/peerj.9604_

## Round 0.1 · original submission · Major Revisions

The two reviewers agree that the MS is interesting. R1 urges some re-analysis to fully test the impact of outgroups (vs. previous studies'), which I agree is important. Otherwise the recommended changes are mainly wording, as are R2's-- which are phrased purely as suggestions but I think merit inclusion in the revised MS unless the author sees a strong arguing point against that. In any case please address all of these issues individually in a clear Rebuttal document. Thank you.

Reviewer 1 ·

Basic reporting

no comment

Experimental design

no comment

Validity of the findings

no comment

Additional comments

The MS by Baron brings interesting data and deserves to be published in PeerJ if the author manages to reply to (accepting or comprehensively contesting) the points raised below.

General (major) comments
Lines 170-175: after the first paragraphs of the “Introduction”, where the author discusses several pterosaur phylogenies, he focus on those based on Britt et al. (2018), questioning the choice of outgroups of that data-base. This is fine, but it would be nice to see here some comments on the outgroup choice of the other studies mentioned before. At least as a counterpoint to that of Britt et al. (2018). This would also give the reader the notion that, if the other data-matrices do not have that problem, but are still disagreeing with one another, the reason for that disagreement is probably not outgroup choice. Also, if the problems is the outgroup choice of Britt et al. (2018) one would expect the outcome of running that database with more outgroups would be closer to those of studies with adequate outgroup choice.
Connected to the comment above, I cannot see Herrerasaurus in the phylogeny of Figure 1a-b.
Lines 246-258: the author conducts a series of analyses with the dataset he modified (including both characters and outgroup taxa) from Dalla Vechia (2019), partially with the aim of finding if outgroup choice has an impact on that study. Therefore, in order to do that, it is chief to do a first run (and latter discuss the results) of the analysis with the added outgroup taxa, but without the added characters. In this way, the author can be sure that the new results he is getting is actually from the new outgroup choice, and not from the characters he added.
Lines 271-332: here the author compares his results of running a data matrix modified from Dalla Vechia (2019) with results of other pterosaur phylogenies. This causes confusion, because it is harder for one to see (among the various comparisons) the actual differences between the new results and those of Dalla Vechia (2019). The author should first discuss these differences (between his and Dalla Vechia’s studies) only, and after provide a more generalist comparison with other phylogenies.
Lines 364-366: as Pterosauria Kellner (2003) is node-based and Pterosauromorpha Padian (1997) is stem based. As such (and because the use different specifies), these can never take objective “precedence” over one another, only have the same inclusivity under a given phylogenetic hypothesis.
Lines 373-469: why are new names given to clades “7” and “8” and not to other unnamed clades. In fact, I would be more comfortable if a pterosaur specialist (which is not my case) also reviews this MS, particularly the adequateness of such new names. It is surprizing, for example, that one of the named clades “is the least consistently supported” (line 410)
Line 473: it is not clear to me what is this “resampling” about. Is that the Ezcurra (2016) protocol, right? He uses resampling for the bootstrap calculations. Are you using a bootstrap tree in Fig. 3a?
Lines 471-512: the author comments on the outcomes of two different analytical procedures applied to the same dataset. I see two problems here: (1) why are the results of the first analysis discussed in so much more detail (lines 264-469) that the others? (2) if the focus is on the different procedures of these two analyses, the discussion (here and in the “Discussion” section) also has to be focused on these procedures, not only in the results, e.g., why use implied weights? why the topology changes because of that? It is not enough to compare the results here, you have to investigate the reasons behind the different results. Another option, surely easier and with no significant loss, is to exclude the two last analyses (lines 471-512).

Specific (minor) comments
“Andres” is misspelled in parts.
“but” (I guess) is misspelled as “bar” in parts.
All clades are monophyletic, please do not use “monophyletic clade”.
Line 415: “that” not “the”
Lines 418: choose another word than “subsequently”, because one of such works is older and the other of the same year.
Line 422: you use derived under ‘’ here. This is OK, and should also be used in other parts discussing the position of taxa in trees.
Lines 433-442: a ten-pages etymology is a bit too much.
Line 474: for the sake of completeness, you could mention the position of Dinosauriformes here as well.

Reviewer 2 ·

Basic reporting

No particular issue here. The paper revisits the question of pterosaur relationships, with special attention to basal taxa, including new closely related species to Pterosauria.

Experimental design

The approach is correct. One always might have different ways, but there is no problem in the way that the authors delimited his study.

Validity of the findings

Conclusions are correct based on the method used.

Additional comments

The ms. by Baron essentially revisits the relationships of pterosaurs, including a several new taxa outside Pterosauria in order to test the results known to date. There are not that many papers that try to rigorously test the relationship in group relationship of basal Pterosauria and therefore I welcome and endorse the publication of this study.

There is are not specific problem with this analysis - one could include other data-matrices or discuss different characters and perhaps use other datasets. But this does not invalidate the interesring approach of the author to this complex problem that surely will have follow up that might show different results.

One point that the author should point out that dinosaurs were included in phylogenetic analysis before, even dinosauromorphs. The information is there in his ms, but I think that he should make this clearer, citing the appropriated contributions, including the first studies that included such taxa.

Another point that might be interesting to clarify is regarding supports. Baron made a comment about low support of Bremer support for some clades of previous studies. But is this not the rule? The fact that there is little support does not mean that the retrieved relationship is not correct. Perhaps he would like to elaborate on that (optional).

Lastly, it is hard to keep up with literature these days. Perhaps, just for the sake of completeness, the author might like to include in the introduction the new discoveries of pterosaurs in the Antarctic Peninsula (optional).

---

## Round 0.2 · accepted · Accept

The reviewer is pleased with the revisions. However, the new Phylocode/Phylonyms means that this paper could be quite influential on future pterosaur taxonomy-- which may be good, but may also incur substantial confusion or instability that may result in impeded communication amongst experts. The other expert reviewer had no specific comments on this issue in the last round. Hence it would be best if the author takes a fresh look at this issue and considers what might "go wrong" if the phylogeny ends up being quite wrong. From my experience in theropod phylogenetic taxonomy I've seen some messes result. Otherwise, the paper seems ready to go-- congratulations!